

# Several methods for assessing research waste in reviews with a systematic search: a scoping review

Louise Olsbro Rosengaard, Mikkel Zola Andersen, Jacob Rosenberg and Siv Fonnes

Center for Perioperative Optimization, Department of Surgery, Copenhagen University Hospital - Herlev and Gentofte, Denmark

## ABSTRACT

**Background**. Research waste is present in all study designs and can have significant consequences for science, including reducing the reliability of research findings and contributing to the inefficient use of resources. Estimates suggest that as much as 85% of all biomedical research is wasted. However, it is uncertain how avoidable research waste is assessed in specific types of study designs and what methods could be used to examine different aspects of research waste. We aimed to investigate which methods, systematic reviews, scoping reviews, and overviews of reviews discussing research waste, have used to assess avoidable research waste.

**Materials and Methods**. We published a protocol in the Open Science Framework prospectively (https://osf.io/2fbp4). We searched PubMed and Embase with a 30-year limit (January 1993–August 2023). The concept examined was how research waste and related synonyms (*e.g.*, unnecessary, redundant, duplicate, *etc.*) were assessed in reviews with a systematic search: systematic, scoping, or overviews of reviews. We extracted data on the method used in the review to examine for research waste and for which study design this method was applied.

**Results**. The search identified 4,285 records of which 93 reviews with systematic searches were included. The reviews examined a median of 90 (range 10–6,781) studies, where the study designs most commonly included were randomized controlled trials (48%) and systematic reviews (33%). In the last ten years, the number of reports assessing research waste has increased. More than 50% of examined reviews reported evaluating methodological research waste among included studies, typically using tools such as one of Cochrane Risk of Bias tools ($n = 8$) for randomized controlled trials or AMSTAR 1 or 2 ($n = 12$) for systematic reviews. One fourth of reviews assessed reporting guideline adherence to *e.g.*, CONSORT ($n = 4$) for randomized controlled trials or PRISMA ($n = 6$) for systematic reviews.

**Conclusion**. Reviews with systematic searches focus on methodological quality and reporting guideline adherence when examining research waste. However, this scoping review revealed that a wide range of tools are used, which may pose difficulties in comparing examinations and performing meta-research. This review aids researchers in selecting methodologies and contributes to the ongoing discourse on optimizing research efficiency.

Corresponding author
Louise Olsbro Rosengaard,
rosengaardlouise@gmail.com

## INTRODUCTION

Research waste refers to practices within research that are inefficient, unnecessary, or fail to deliver reliable results. It is an issue that has been receiving increasing attention in recent years. The MINUS definition divides research waste into five main aspects: Methodological, Invisible, Negligible, Underreported, and Structural research waste (*Rosengaard et al., 2024b*). Each of these aspects addresses different types of research waste that can occur at any stage of a study. Research waste is present in all study designs (*Ioannidis et al., 2014*) and can have significant consequences for science, including reducing the reliability of research findings and contributing to the inefficient use of resources (*Glasziou et al., 2014*). Estimates suggest that as much as 85% of all biomedical research is wasted (*Chalmers & Glasziou, 2009*). Over the past 30 years, the volume of publications in the biomedical sciences has been growing rapidly (*Bastian, Glasziou & Chalmers, 2010*). Systematic evaluations, such as systematic reviews, could help manage this rapid data growth and improve research quality. Systematic reviews are considered a fundamental unit of knowledge translation (*Tricco, Tetzlaff & Moher, 2011*). Systematic reviews utilize a research study design that evaluates and synthesizes the existing literature reporting on a specific research question. This literature is acquired through a systematic search and may potentially include different study designs. Scoping reviews and overviews of reviews also build on systematic searches. Provided that systematic reviews are conducted using thorough methodology, this gives the potential to produce results that are as least biased as possible.

There is uncertainty in how avoidable research waste is assessed in specific types of study designs and what methods could be used to examine different aspects of research waste (*Pussegoda et al., 2017a*). We aimed to investigate which methods systematic reviews, scoping reviews, and overviews of reviews discussing research waste, are used to assess avoidable research waste.

## MATERIALS & METHODS

### Protocol and eligibility criteria

We used a scoping review approach to investigate the methods used in reviews based on a systematic search for papers that examined or discussed research waste, following the guidance for scoping reviews previously established (*Munn et al., 2018*). We reported our study following the Preferred Reporting Items for Systematic reviews and Meta-Analysis extension for Scoping Reviews (PRISMA-ScR) reporting guideline (*Tricco et al., 2018*) and defined the eligibility criteria according to the Joanna Briggs Institute guidance (*Peters et al., 2021*). The protocol was registered at Open Science Framework (*Rosengaard et al., 2023*). This scoping review reports on a prespecified secondary outcome in the protocol, whereas the main outcome is reported elsewhere (*Rosengaard et al., 2024b*).

Utilizing the population, concept, and context framework for scoping reviews (*Peters et al., 2021*), the concept examined was how research waste and related synonyms (*e.g.*, unnecessary, redundant, duplicate, *etc.*) were assessed. We included reviews that employed a systematic search strategy, as a search process that is comprehensive, structured, and replicable, intended to identify all relevant literature on a given topic. This, *e.g.*, includes systematic reviews, scoping reviews, and overviews of reviews, all of which must have a clearly documented search strategy to be considered for inclusion. Reviews were included if they defined or discussed research waste or its synonyms as part of their focus. The context of our scoping review related to any method reported to be used to examine research waste in their included reviews for different study designs. We included reviews published in scholarly journals with systematic searches in all languages from the last 30 years (1993–2023) to provide a contemporary view of the subject. We chose a 30-year timeframe to review recent advances in research waste assessment methods. We excluded reports that did not include a definition or discussion of research waste or related terms or synonyms. Furthermore, we excluded reviews that did not directly examine research waste, reviews of veterinary sciences, reviews that did not analyze health-related research on humans, and titles marked as retracted.

## Information sources and search

Through a discussion of examples of research waste in the author group, several ways to describe the concept were agreed upon. Pilot searches were conducted to find synonyms of research waste based on these wordings. The final search string was developed in PubMed and adapted for Embase. It was consulted with and approved by an information specialist. The last date of the search was August 17, 2023. The full search strategy is presented in the protocol (*Rosengaard et al., 2023*). We performed a backward citation search (*Greenhalgh & Peacock, 2005*), and the final included reviews were crosschecked in the Retraction Watch database (*Crossref, 2024*).

## Selection and data

Two reviewers independently screened records in Covidence (*Veritas Health Innovation, 2024*), initially on title and abstract and subsequently in full text according to the eligibility criteria. Conflicts were resolved through consensus discussions. Data extraction was performed independently in duplicate. Data were extracted to pilot forms in Microsoft Excel (Microsoft Corporation, Redmond, WA, USA). We extracted the following data from the included reviews: type of review with a systematic search (systematic review, scoping review, or overview of reviews), number of included studies in the reviews, the study designs that the reviews included, the aim of the reviews, which aspects of research waste that were examined according to the MINUS aspect framework (*Rosengaard et al., 2024b*), and the methods reported by authors as used for investigating research waste. During data extraction, we retrospectively assigned each research waste assessment method to a waste category defined by the MINUS framework (*Rosengaard et al., 2024b*).

The aspects of research waste as defined by MINUS are Methodological, Invisible, Negligible, Underreported, and Structural research waste (*Rosengaard et al., 2024b*).

Methodological research waste includes waste related to study conduct, methodological quality, and recruitment and retention of patients. Invisible research waste concerns a lack of data sharing, non-publication of conducted work, inaccessible research, and discontinuation of trials. Negligible research waste includes unnecessary duplication of research work and unjustified research, *e.g.*, lack of prior literature search. Underreported research waste especially concerns reporting guideline adherence, written reporting, and heterogeneous outcome reporting. Finally, structural research waste concerns prioritization in research, implementation in clinical settings, inequity in health, patient involvement, management, and collaboration. We synthesized the results in a descriptive analysis, and the tools used in the included reviews are presented in tables and figures. We utilized an UpSet plot (*Conway, Lex & Gehlenborg, 2017*) to visualize overlapping aspects when examining research waste as the five aspects of MINUS (*Rosengaard et al., 2024b*). We categorized the different tools used to evaluate research waste according to which study designs the reviews included in their examination (*Pollock et al., 2023*). All extracted data and excluded studies are shared (*Rosengaard et al., 2024a*).

## RESULTS

### Selection and characteristics

The study selection process is presented in Fig. 1. The search identified 4,285 records of which 93 reviews were included in this scoping review (*Pussegoda et al., 2017a*; *Clyne et al., 2020*; *Ker & Roberts, 2015*; *Ndounga Diakou et al., 2017*; *Sheth et al., 2011*; *Wu et al., 2022*; *Arundel & Mott, 2023*; *Kostalova et al., 2022*; *Limones et al., 2022*; *Doumouchtsis et al., 2019*; *Habre et al., 2014*; *Webbe et al., 2020*; *Pergialiotis et al., 2018*; *Créquit et al., 2016*; *Hacke & Nunan, 2019*; *Martel et al., 2012*; *Pussegoda et al., 2017b*; *Reddy et al., 2023*; *Slattery, Saeri & Bragge, 2020*; *Townsend et al., 2019*; *Synnot et al., 2018*; *Choi et al., 2022*; *Whear et al., 2022*; *Bendersky et al., 2023*; *Frost et al., 2018*; *Chambers et al., 2014*; *Hancock & Mattick, 2020*; *Houghton et al., 2020*; *Collins & Lang, 2018*; *Patarčić et al., 2015*; *Pandis et al., 2021*; *Fisher et al., 2022*; *Bolland, Avenell & Grey, 2018*; *Ramke et al., 2018*; *Sebastianski et al., 2019*; *Amad et al., 2019*; *Clarke, Brice & Chalmers, 2014*; *Avau et al., 2023*; *Xu et al., 2021*; *Mikelis & Koletsi, 2022*; *Ahmed Ali et al., 2018*; *Papathanasiou et al., 2016*; *Mercieca-Bebber et al., 2016*; *Gale et al., 2013*; *Shepard et al., 2023*; *Wright et al., 2000*; *Briel et al., 2016*; *Grégory et al., 2020*; *Sauzet, Kleine & Williams, 2016*; *Morgan et al., 2021*; *Siemens et al., 2022*; *Hey et al., 2017*; *Palmer et al., 2018*; *Blanco-Silvente et al., 2019*; *Cook, 2014*; *Maxwell et al., 2023*; *Sawin & Robinson, 2016*; *Lund et al., 2022*; *Sharma et al., 2019*; *Bolland, Grey & Avenell, 2018*; *Johnson et al., 2020*; *Torgerson et al., 2020*; *Gysling, Khan & Caruana, 2023*; *McGill et al., 2020*; *Agbadjé et al., 2022*; *Meneses-Echavez et al., 2019*; *Page et al., 2016*; *Mercieca-Bebber et al., 2022*; *Andaur Navarro et al., 2022*; *Dhiman et al., 2021*; *Okomo et al., 2019*; *Feng et al., 2022*; *Yu et al., 2018*; *Harman et al., 2021*; *Pascoe et al., 2021*; *Duffy et al., 2017*; *Evans et al., 2020*; *Velde et al., 2021*; *Rives-Lange et al., 2022*; *Cirkovic et al., 2020*; *Butcher et al., 2020*; *Dal Santo et al., 2023*; *Bero, Chiu & Grundy, 2019*; *Boutron & Ravaud, 2018*; *Klaic et al., 2022*; *Holmes et al., 2020*; *Cruz Rivera et al., 2017*; *Levati et al., 2016*; *Albarqouni, Elessi & Abu-Rmeileh, 2018*; *Bentley et al., 2019*; *Coffey et al., 2022*; *Tybor et al., 2018*; *Nankervis,*

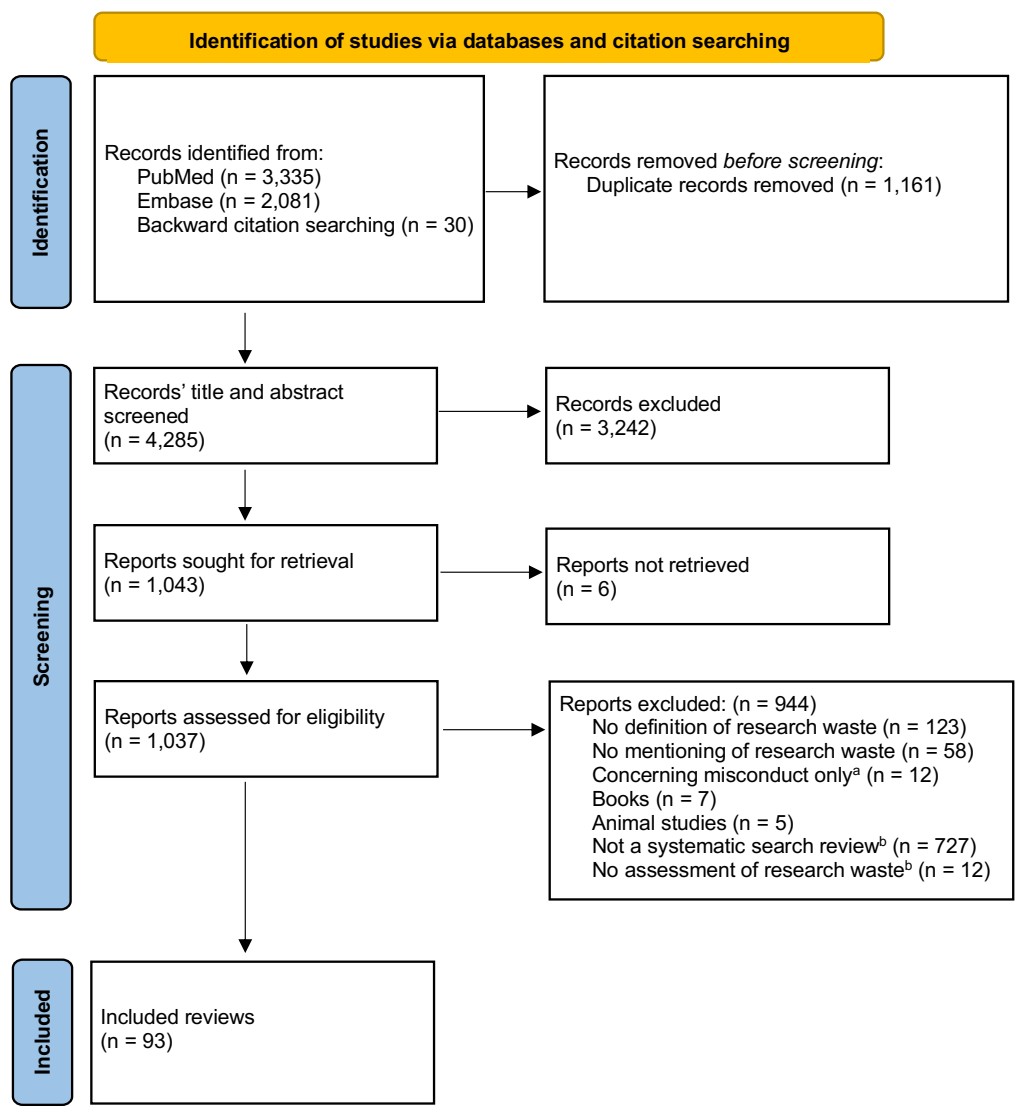

**Figure 1** **PRISMA flow diagram depicting the selection of reports in the scoping review.** (A) Research misconduct involves intentional violations of ethical standards and integrity, fundamentally different from research waste, *e.g.*, inefficiencies and inadequacies in the research process that lead to the loss of valuable resources, duplication of effort, and reduced reliability of findings. (B) Reported elsewhere (*Rosengaard et al., 2024b*).

*Maplethorpe & Williams, 2011*). Characteristics of the included, reviews are presented in Table 1. Of the included reviews, the majority were systematic reviews ($n = 73$). The reviews examined a median of 90 (10–6,781) studies for research waste. The included reviews' most frequently examined study designs were randomized controlled trials (RCTs) (47%) and systematic reviews or meta-analyses (33%) (Table 1). About one-fourth (23%) of the reports included multiple types of study design. In the last ten years, the number of reports assessing research waste has increased (Table 1).

**Table 1** Characteristics of included reviews (*n* = 93) given as numbers (%).

| Characteristics of reviews | |
| --- | --- |
| Type of report | |
|     Systematic reviews | 73 (78) |
|     Overviews of reviews | 12 (13) |
|     Scoping reviews | 8 (9) |
| Type of study design included | |
|     Randomized controlled trials | 44 (47) |
|     Systematic reviews, meta-analyses, scoping reviews | 31 (33) |
|     Observational studies[a] | 15 (16) |
|     Non-randomized controlled trials | 14 (15) |
|     Other[b] | 13 (14) |
| Year of publication | |
|     1993–2002 | 1 (1) |
|     2003–2012 | 3 (3) |
|     2013–2017 | 21 (22) |
|     2018–2023 | 68 (73) |

**Notes.**

Several reviews (23%) included multiple study designs.

[a] Including cohort (*n* = 9), prognostic and prediction (*n* = 2), cross-sectional (*n* = 3), longitudinal (*n* = 2), and case-control studies (*n* = 1).

[b] Empirical studies (*n* = 1), guidelines (*n* = 1), expert opinion (*n* = 1), meta-research (*n* = 3), qualitative studies (*n* = 2), surveys (*n* = 1), protocols (*n* = 2), and case-reports (*n* = 2).

## Aspects of research waste being examined

Figure 2 shows which MINUS aspects of research waste that was examined by the included reviews, and that about half of the reviews examine multiple aspects of research waste. Methodological research waste was the most assessed aspect (*n* = 51). Here, reviews mainly focused on the assessment of conduct examination and methodological quality assessment (Fig. 3). Conduct examination refers to assessing how the research was conducted and identifying potential practices that may have led to waste. Quality assessment refers to evaluating the methodological quality or rigor of the included reviews, focusing on how well the studies were conducted and reported. Underreported (*n* = 42) and structural (*n* = 22) research waste were also commonly investigated aspects (Fig. 2).

## Across the included study designs

Figure 3 shows each aspect of MINUS and what the reviews examined within these aspects. When examining each research waste aspect of MINUS, regardless of study design, they included the following evaluations.

Methodological research waste was evaluated through quality assessment (32%) or conduct examination (21%) (Fig. 3). When assessing methodological quality, specific tools were used, primarily the risk of bias tools that are specific to the design of the review's included studies (Table 2). Two reports used self-constructed risk of bias tools to assess for quality assessments (*Collins & Lang, 2018*; *Patarčić et al., 2015*).

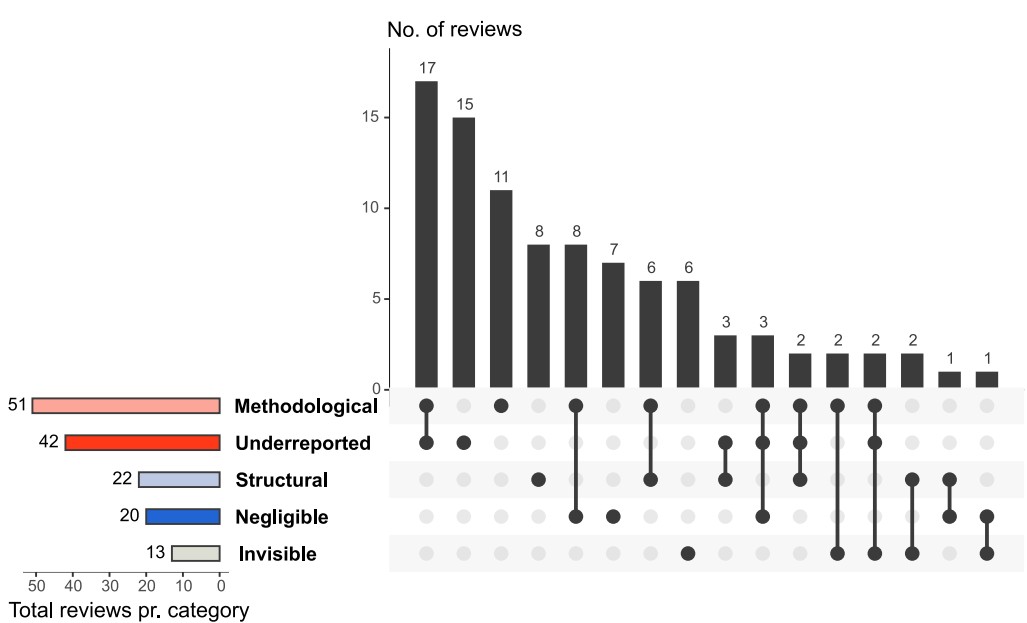

**Figure 2** An UpSet plot (*Greenhalgh & Peacock, 2005*) of the distribution of the examined aspect of research waste according to MINUS and the overlap of aspects examined by the included reviews ($n = 93$).

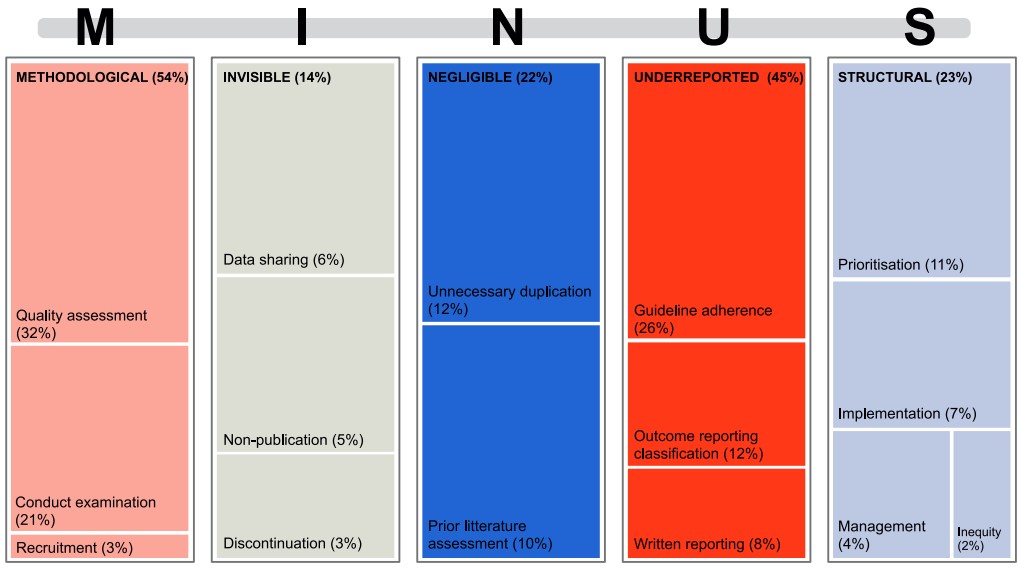

**Figure 3** Treemap for each of the examined five aspects of research waste MINUS (*Rosengaard et al., 2024b*) and what the reviews investigated within each aspect.

Invisible research waste was mainly evaluated concerning inadequate data sharing (6%), non-publication (5%), and discontinuation (3%) (Fig. 3). Only one review (*Mercieca-Bebber et al., 2016*) used a tool to examine invisible research, *i.e.,* the Framework Method for

**Table 2  Methods to assess for methodological research waste according to studies examined.**

| The examined study design | Methodological (n = 51) | | |
|---|---|---|---|
| | Quality assessment (reference to the method) | n | Reference to the review utilizing the method |
| Randomized controlled trials | Cochrane RoB 1 (*Higgins, Altman & Sterne, 2017*) | 5 | *Clyne et al. (2020)*; *Ker & Roberts (2015)*; *Ndounga Diakou et al. (2017)*; *Sheth et al. (2011)*; *Wu et al. (2022)* |
| | Cochrane RoB 2 (*Sterne et al., 2019*) | 3 | *Arundel & Mott (2023)*; *Kostalova et al. (2022)*; *Limones et al. (2022)* |
| | Jadad (*Jadad et al., 1996*) | 4 | *Créquit et al. (2016)*; *Hacke & Nunan (2019)*; *Martel et al. (2012)*; *Pussegoda et al. (2017b)* |
| | GRADE (*Atkins et al., 2004*) | 1 | *Habre et al. (2014)* |
| Systematic reviews, meta-analysis | AMSTAR 1 (*Shea et al., 2007*) | 9 | *Pussegoda et al. (2017a)*; *Créquit et al. (2016)*; *Hacke & Nunan (2019)*; *Martel et al. (2012)*; *Pussegoda et al. (2017b)*; *Reddy et al. (2023)*; *Slattery, Saeri & Bragge (2020)*; *Townsend et al. (2019)*; *Synnot et al. (2018)* |
| | OQAQ (*Oxman & Guyatt, 1991*) | 4 | *Pussegoda et al. (2017a)*; *Sheth et al. (2011)*; *Pussegoda et al. (2017b)*; *Reddy et al. (2023)* |
| | AMSTAR 2 (*Shea et al., 2017*) | 3 | *Choi et al. (2022)*; *Whear et al. (2022)*; *Bendersky et al. (2023)* |
| | GRADE (*Atkins et al., 2004*) | 2 | *Pussegoda et al. (2017a)*; *Frost et al. (2018)* |
| | QUIPS (*Hayden et al., 2013*) | 1 | *Townsend et al. (2019)* |
| | ROBIS (*Whiting et al., 2015*) | 1 | *Frost et al. (2018)* |
| | Jadad (*Jadad et al., 1996*) | 1 | *Reddy et al. (2023)* |
| | DARE (*Centre for Reviews and Dissemination, 2024*) | 1 | *Chambers et al. (2014)* |
| | Mulrow (*Mulrow, 1987*) | 1 | *Pussegoda et al. (2017b)* |
| | Sacks (*Sacks et al., 1987*) | 1 | *Pussegoda et al. (2017b)* |
| | Oxford levels of evidence (*Wright, Swiontkowski & Heckman, 2003*) | 1 | *Reddy et al. (2023)* |
| Observational studies | Newcastle-Ottawa scale (*Wells et al., 2021*) | 1 | *Sheth et al. (2011)* |
| Medical education studies | MERSQI (*Reed et al., 2007*) | 1 | *Hancock & Mattick (2020)* |
| Qualitative studies | CASP (*Critical Appraisal Skills Programme,* ) | 1 | *Houghton et al. (2020)* |

Notes.
AMSTAR, A MeaSurement Tool to Assess systematic Reviews; RoB, Risk of Bias; CASP, Critical Appraisal Skills Programme; DARE, The Database of Abstracts of reviews of Effects; GRADE, Grading of Recommendations Assessment, Development and Evaluation.; Jadad, a tool developed by Alex Jadad; MERSQI, The Medical Education Research Study Quality Instrument; n, number; OQAQ, Overview Quality Assessment Questionnaire; QUIPS, Quality in Prognosis Studies; ROBIS, Risk of Bias in Systematic Reviews.

analyzing qualitative data (*Gale et al., 2013*) to account for missing data. Non-publication was primarily examined by checking the publication status of reports in registries such as ClinicalTrials.gov (*Shepard et al., 2023*).

Negligible research waste was assessed by examining unnecessary duplication of research effort (12%) or by reviewing the included reports for whether they had adequately justified conducting their research (10%) (Fig. 3 and Table 3), *e.g.*, by checking the introduction for assessment of prior literature within the topic (*Ker & Roberts, 2015*; *Sheth et al., 2011*; *Habre et al., 2014*; *Martel et al., 2012*; *Chambers et al., 2014*; *Mikelis & Koletsi, 2022*; *Bolland, Grey & Avenell, 2018*; *Johnson et al., 2020*; *Torgerson et al., 2020*).

**Table 3  Methods to assess for negligible research waste according to studies examined.**

| The examined study design | Negligible (n = 20) | | |
|---|---|---|---|
| | Unnecessary duplication [reference to the method] | n | Reference to the review utilizing the method |
| Randomized controlled trials | Meta-analysis (*Deeks, Higgins & Altman, 2023*) | 3 | *Ker & Roberts (2015)*; *Bolland, Avenell & Grey (2018)*; *Blanco-Silvente et al. (2019)*; *Cook (2014)* |
| | Trial sequential analysis (*Thorlund et al., 2017*) | 2 | *Ker & Roberts (2015)*; *Blanco-Silvente et al. (2019)* |
| Systematic reviews, meta-analysis, scoping reviews | Sankey diagrams | 2 | *Whear et al. (2022)*; *Maxwell et al. (2023)* |
| | Network analysis (*Yan & Ding, 2009*) | 1 | *Whear et al. (2022)* |
| | RPRCI and RSSCI (*Robinson & Goodman, 2011*) | 1 | *Sawin & Robinson (2016)* |
| | CCA (*Pieper et al., 2014*) | 1 | *Reddy et al. (2023)* |

Notes.

n, number; CCA, Corrected covered area; RPRCI, Robison's Prior Research Citation Index; RSSCI, Robinson's Sample Size Citation Index.

Underreported research waste was examined either through reporting guideline adherence (26%), heterogeneous outcome reporting (12%), or the written reporting (8%) (Fig. 3). The most common reporting guidelines utilized are displayed in Table 4. Reporting of outcomes was primarily examined when creating a core outcome set.

Structural research waste was mainly evaluated by examining prioritization (11%) or implementation (7%) (Fig. 3). When examining prioritization in the research field, evidence maps were commonly used to create an overview of the subject and where future research should be prioritized (*Synnot et al., 2018*; *Choi et al., 2022*; *Bendersky et al., 2023*; *Avau et al., 2023*; *Rives-Lange et al., 2022*; *Coffey et al., 2022*; *Tybor et al., 2018*; *Nankervis, Maplethorpe & Williams, 2011*). Implementation could be measured by implementation frameworks, where one report found 24 different implementation frameworks used to measure research impact (*Cruz Rivera et al., 2017*). The included reviews used different frameworks, but for the mixed study designs the Conceptual Framework for Implementation Fidelity (CFIF) (*Agbadjé et al., 2022*; *Holmes et al., 2020*) and the related Conceptual Framework for Implementation Research (CFIR) (*Holmes et al., 2020*) were used (Table 5). Thirty-three reports used a descriptive analysis to examine for research waste. These are categorized in Table 6 (*Clyne et al., 2020*; *Limones et al., 2022*; *Habre et al., 2014*; *Slattery, Saeri & Bragge, 2020*; *Pandis et al., 2021*; *Fisher et al., 2022*; *Bolland, Avenell & Grey, 2018*; *Ramke et al., 2018*; *Sebastianski et al., 2019*; *Clarke, Brice & Chalmers, 2014*; *Papathanasiou et al., 2016*; *Briel et al., 2016*; *Grégory et al., 2020*; *Sauzet, Kleine & Williams, 2016*; *Morgan et al., 2021*; *Siemens et al., 2022*; *Hey et al., 2017*; *Palmer et al., 2018*; *Lund et al., 2022*; *Sharma et al., 2019*; *Pascoe et al., 2021*; *Duffy et al., 2017*; *Evans et al., 2020*; *Velde et al., 2021*; *Rives-Lange et al., 2022*; *Cirkovic et al., 2020*; *Butcher et al., 2020*; *Dal Santo et al., 2023*; *Bero, Chiu & Grundy, 2019*; *Boutron & Ravaud, 2018*; *Cruz Rivera et al., 2017*; *Levati et al., 2016*; *Albarqouni, Elessi & Abu-Rmeileh, 2018*; *Bentley et al., 2019*).

**Table 4  Methods to assess for underreported research waste according to studies examined.**

| The examined study design | Underreported (n = 42) | | | | | |
|---|---|---|---|---|---|---|
| | Reporting guideline adherence (reference to the method) | n | Reference to the review utilizing the method | Reporting of outcomes [reference to the method] | n | Reference to the review utilizing the method |
| Randomised controlled trials | CONSORT (*Hopewell et al., 2008*) | 4 | *Ndounga Diakou et al. (2017)*; *Limones et al. (2022)*; *Gysling, Khan & Caruana (2023)*; *McGill et al. (2020)* | MOMENT (*Harman et al., 2013*) | 2 | *Doumouchtsis et al. (2019)*; *Pergialiotis et al., (2018)* |
| | TIDieR (*Hoffmann et al., 2014*) | 3 | *Ndounga Diakou et al. (2017)*; *Agbadjé et al. (2022)*; *Meneses-Echavez et al. (2019)* | ORBIT (*Kirkham et al., 2010*) | 1 | *Webbe et al. (2020)* |
| Systematic reviews, meta-analysis, scoping reviews | PRISMA (*Hutton et al., 2015*) | 6 | *Pussegoda et al. (2017a)*; *Hacke & Nunan (2019)*; *Pussegoda et al. (2017b)*; *Whear et al. (2022)*; *Fisher et al. (2022)*; *Page et al. (2016)* | | | |
| | QUOROM (*Moher et al., 1999*) | 4 | *Créquit et al. (2016)*; *Pussegoda et al. (2017b)*; *Reddy et al. (2023)* | | | |
| | TIDieR (*Hoffmann et al., 2014*) | 1 | *Frost et al. (2018)* | | | |
| | CONSORT-PRO (*Calvert et al., 2013*) | 1 | *Mercieca-Bebber et al. (2022)* | | | |
| Observational studies | TRIPOD (*Collins et al., 2015*) | 2 | *Andaur Navarro et al. (2022)*; *Dhiman et al. (2021)* | | | |
| | STROBE (*Von Elm et al., 2007*) | 1 | *Okomo et al. (2019)* | | | |
| | STARD (*Bossuyt et al., 2015*) | 1 | *Feng et al. (2022)* | | | |
| Non-randomised controlled trials | TIDieR (*Hoffmann et al., 2014*) | 2 | *Agbadjé et al. (2022)*; *Yu et al. (2018)* | COMET (*Dodd et al., 2018*) | 1 | *Harman et al. (2021)* |
| | ARRIVE (*Dodd et al., 2018*; *Kilkenny et al., 2014*) | 1 | *Collins & Lang (2018)* | | | |

**Notes.**

N, number; COMET, Core Outcome Measures in Effectiveness Trials; CONSORT, Consolidated Standards of Reporting Trials; MOMENT, Management of Otitis Media with Effusion in Cleft Palate; n, number; ORBIT, Outcome Reporting Bias In Trials; PRISMA, Preferred Reporting Items for Systematic Reviews and Meta-Analyses; QUOROM, Quality of Reporting of Metaanalyses; STARD, Standards for Reporting Diagnostic Accuracy; STROBE, The Strengthening the Reporting of Observational Studies in Epidemiology; TIDieR, Template for Intervention Description and Replication; TRIPOD, Transparent Reporting of a multivariable prediction model for Individual Prognosis or Diagnosis.

**Table 5** Methods to assess for structural research waste according to studies examined.

| The examined study design | Structural ($n = 22$) | | |
| --- | --- | --- | --- |
| | Implementation methods [reference to the method] | n | Reference to the review utilizing the method |
| Randomized controlled trials | The ten adapted Peters criteria (*Peters et al., 2013*) | 1 | *Kostalova et al. (2022)* |
| Systematic reviews, meta-analysis, scoping reviews | Framework of implementability (*Klaic et al., 2022*) | 1 | *Klaic et al. (2022)* |
| Mixed | CFIF (*Carroll et al., 2007*) | 2 | *Agbadjé et al. (2022)*; *Holmes et al. (2020)* |
| | CFIR (*Damschroder et al., 2009*) | 1 | *Holmes et al. (2020)* |

Notes.
N, number; CFIF, Conceptual Framework for Implementation Fidelity; CFIR, Conceptual Framework for Implementation Research.

**Table 6** Reviews included a descriptive analysis of research waste grouped by included study design and MINUS.

| The examined study design | Methodological | Invisible | Negligible | Under-reported | Structural |
| --- | --- | --- | --- | --- | --- |
| Randomized controlled trials | *Pandis et al. (2021)* | *Briel et al. (2016)*; *Grégory et al. (2020)*; *Sauzet, Kleine & Williams (2016)*; *Morgan et al. (2021)*; *Siemens et al. (2022)*; *Hey et al. (2017)*; *Palmer et al. (2018)* | | *Clyne et al. (2020)*; *Limones et al. (2022)*; *Pandis et al. (2021)*; *Papathanasiou et al. (2016)*; *Pascoe et al. (2021)*; *Duffy et al. (2017)*; *Evans et al. (2020)*; *Velde et al. (2021)*; *Rives-Lange et al. (2022)* | *Grégory et al. (2020)* |
| Systematic reviews | *Fisher et al. (2022)* | *Siemens et al. (2022)* | *Clarke, Brice & Chalmers (2014)*; *Lund et al. (2022)* | | |
| Observational studies | | | | *Cirkovic et al. (2020)* | |
| Non-randomized controlled trials | | *Hey et al. (2017)* | | | |
| Mixed | *Bolland, Avenell & Grey (2018)*; *Ramke et al. (2018)*; *Sebastianski et al. (2019)* | *Ramke et al. (2018)*; *Palmer et al. (2018)* | *Bolland, Avenell & Grey (2018)*; *Lund et al. (2022)*; *Sharma et al. (2019)* | *Butcher et al. (2020)*; *Dal Santo et al. (2023)*; *Bero, Chiu & Grundy (2019)*; *Boutron & Ravaud (2018)* | *Slattery, Saeri & Bragge (2020)*; *Sharma et al. (2019)*; *Cruz Rivera et al. (2017)*; *Levati et al. (2016)*; *Albarqouni, Elessi & Abu-Rmeileh (2018)*; *Bentley et al. (2019)* |

## Study design and tools to assess research waste
### Randomized controlled trials

A total of 44 reviews assessed randomized controlled trials. Methodological research waste assessment focused on three issues: sample size or power analysis, quality, and trial recruitment. Sample size or power analysis tests were performed or repeated by two reports (*Ahmed Ali et al., 2018*; *Papathanasiou et al., 2016*). Quality assessments were performed through bias assessment, especially Cochrane risk of bias tool 1 (*Clyne et al., 2020*; *Ker & Roberts, 2015*; *Ndounga Diakou et al., 2017*; *Sheth et al., 2011*; *Wu et al., 2022*) and 2 (*Arundel & Mott, 2023*; *Kostalova et al., 2022*; *Limones et al., 2022*), Table 2. Other tools for quality assessment applied included the Jadad scale (*Doumouchtsis et al., 2019*; *Habre et al., 2014*; *Webbe et al., 2020*; *Pergialiotis et al., 2018*) or Grading of Recommendations

Assessment, Development and Evaluation (GRADE) (*Arundel & Mott, 2023*). Recruitment to trials was assessed descriptively (*Houghton et al., 2020*; *Briel et al., 2016*) or through a statistical test to compare different factors associated with recruitment (*McGill et al., 2020*). Invisible research waste was evaluated by checking ClinicalTrials.gov for registered trials and if they resulted in a publication (*Shepard et al., 2023*) or by Egger's test for publication bias (*Wright et al., 2000*). When testing for negligible research, meta-analyses (*Ker & Roberts, 2015*; *Bolland, Avenell & Grey, 2018*; *Blanco-Silvente et al., 2019*; *Cook, 2014*), and Trial Sequential Analyses (*Ker & Roberts, 2015*; *Blanco-Silvente et al., 2019*) were used (Table 3). Underreported research waste was primarily assessed for reporting guideline adherence to either the Consolidated Standards of Reporting Trials (CONSORT) checklist (*Ndounga Diakou et al., 2017*; *Limones et al., 2022*; *Gysling, Khan & Caruana, 2023*; *McGill et al., 2020*) or the Template for Intervention Description and Replication (TIDieR) checklist (*Ndounga Diakou et al., 2017*; *Agbadjé et al., 2022*; *Gysling, Khan & Caruana, 2023*), Table 4. When classifying and categorizing outcomes, two methods were applied: Management of Otitis Media with Effusion in Cleft Palate (MOMENT) (*Doumouchtsis et al., 2019*; *Pergialiotis et al., 2018*) and Outcome Reporting Bias In Trials (ORBIT) (*Webbe et al., 2020*) (Table 4). In structural research waste, the implementation of research was evaluated by the ten adapted Peters criteria (*Kostalova et al., 2022*) in trials (Table 5).

## Systematic reviews and meta-analysis

A total of 31 reviews focused on systematic reviews with or without meta-analyses. Methodological research waste was evaluated by examination of conduct and quality assessment. Examination of conduct was done by analyzing power (*Amad et al., 2019*; *Clarke, Brice & Chalmers, 2014*), search and screening (*Avau et al., 2023*), statistical analyses (*Fisher et al., 2022*; *Xu et al., 2021*), and whether the reviews had published a protocol (*Mikelis & Koletsi, 2022*). Tools for quality assessment are displayed in Table 2 along with their count. The most used tools for quality assessment were A MeaSurement Tool to Assess systematic Reviews (AMSTAR) 1 (*Créquit et al., 2016*; *Hacke & Nunan, 2019*; *Martel et al., 2012*; *Pussegoda et al., 2017b*; *Reddy et al., 2023*; *Slattery, Saeri & Bragge, 2020*; *Townsend et al., 2019*; *Synnot et al., 2018*), AMSTAR 2 (*Choi et al., 2022*; *Whear et al., 2022*; *Bendersky et al., 2023*), and Overview Quality Assessment Questionnaire (OQAQ) (*Sheth et al., 2011*; *Pussegoda et al., 2017b*; *Reddy et al., 2023*) (Table 4). Assessment of negligible research waste in systematic reviews was either focused on prior literature assessment or by exploring overlap among different reviews. Prior literature assessment was evaluated by Robinson's Prior Research Citation Index (RPRCI) (*Sawin & Robinson, 2016*) or Robinson's Sample Size Citation Index (RSSCI) (*Sawin & Robinson, 2016*). Overlap could be calculated using the Corrected Covered Area (CCA) (*Reddy et al., 2023*), by network analysis (*Whear et al., 2022*), or with a Sankey diagram (*Maxwell et al., 2023*) (Table 3). Reporting guideline adherence among systematic reviews and meta-analyses was primarily checked using the Preferred Reporting Items for Systematic Reviews and Meta-Analyses (PRISMA) guideline (*Pussegoda et al., 2017a*; *Hacke & Nunan, 2019*; *Pussegoda et al., 2017b*; *Whear et al., 2022*; *Fisher et al., 2022*; *Page et al., 2016*) or Quality of Reporting of Meta-analyses (QUOROM) (*Hacke & Nunan, 2019*; *Pussegoda et al., 2017b*; *Reddy et al., 2023*). Other checklists used

are listed in Table 4. One report examined guideline documentation (*Mikelis & Koletsi, 2022*) and another examined protocol adherence (*Page et al., 2016*). The Framework of Implementability (*Klaic et al., 2022*) was used to evaluate the impact of the review when published (Table 5).

### Observational studies

A total of 15 reviews focused on observational studies. Methodological quality among observational studies was evaluated using the Newcastle-Ottawa scale risk of bias tool (*Kostalova et al., 2022*) (Table 2) or by the conduct of statistical tests (*Cirkovic et al., 2020*) or sample size calculations (*Feng et al., 2022*). Underreported research waste was assessed by reporting guideline adherence to the Transparent Reporting of a multivariable prediction model for Individual Prognosis or Diagnosis (TRIPOD) statement (*Andaur Navarro et al., 2022*; *Dhiman et al., 2021*), The STrengthening the Reporting of OBservational studies in Epidemiology (STROBE) guideline (*Page et al., 2016*; *Okomo et al., 2019*), or Standards for Reporting Diagnostic Accuracy (STARD) (*Feng et al., 2022*) (Table 4).

### Non-randomized controlled trials

A total of 14 reviews focused on non-randomized trials. When checking trials for reporting guideline adherence, two reports used the TIDieR checklist (*Agbadjé et al., 2022*; *Yu et al., 2018*) (Table 4). Reporting of outcomes was categorized by the Core Outcome Measures in Effectiveness Trials (COMET) taxonomy (*Harman et al., 2021*). One report used the Accumulating Evidence and Research Organization (AERO) model (*Patarčić et al., 2015*) to prioritize the research subject when planning translational research.

## DISCUSSION

Across all the study designs, over half of the reports examined for methodological research waste, and just under half of the reports examined for underreporting. The reviews examining RCTs primarily used Cochrane Risk of Bias tools for quality assessment and either CONSORT or TIDieR for reporting guideline adherence. Reviews examining systematic reviews focused on negligible research waste and examined for unnecessary duplication. In quality assessment and reporting guideline adherence, the most frequently used tools were AMSTAR and PRISMA, respectively. This review revealed a wide range of tools, which may pose difficulties in comparing examinations and performing meta-research.

As this scoping review only includes reviews with a systematic search, it bears a close resemblance to an overview of reviews. However, our goal was to identify types of methods and explore research waste examinations, so we utilized a scoping review design. Earlier research waste assessments have focused on one or two aspects, while this review addresses all kinds of avoidable research waste (*Pussegoda et al., 2017a*; *Pandis et al., 2021*; *Amad et al., 2019*). This review has limitations as well. In our search strategy, we applied a 30-year limit. Despite this, only two reviews were included from the first 15 years for this scoping review, making it unlikely that much literature was missed because of the time restriction. We only examined those reports that defined research waste as a problem, which could

potentially mean that many articles may address the issue of research waste without explicitly using the terms or synonyms. Thus, they were not included in this review. The protocol was imprecise regarding this research question, but piloted forms were developed before data extraction. In our scoping review, we found various methods for assessing research waste. However, it is important to note that studies with a high risk of bias or questionable research practices may still provide meaningful findings to the scientific community. Therefore, when assessing research waste, we should consider the context and potential value of the study's findings, instead of only focusing on its methodological rigor.

This review provides a unique overview of tools used to examine research waste and may inspire and guide future systematic reviews, scoping reviews, and overviews of reviews. A large portion of the reviews performed a descriptive analysis of the aspects of research waste they wanted to examine (*Clyne et al., 2020*; *Limones et al., 2022*; *Habre et al., 2014*; *Slattery, Saeri & Bragge, 2020*; *Pandis et al., 2021*; *Fisher et al., 2022*; *Bolland, Avenell & Grey, 2018*; *Ramke et al., 2018*; *Sebastianski et al., 2019*; *Clarke, Brice & Chalmers, 2014*; *Papathanasiou et al., 2016*; *Briel et al., 2016*; *Grégory et al., 2020*; *Sauzet, Kleine & Williams, 2016*; *Morgan et al., 2021*; *Siemens et al., 2022*; *Hey et al., 2017*; *Palmer et al., 2018*; *Lund et al., 2022*; *Sharma et al., 2019*; *Pascoe et al., 2021*; *Duffy et al., 2017*; *Evans et al., 2020*; *Velde et al., 2021*; *Rives-Lange et al., 2022*; *Cirkovic et al., 2020*; *Butcher et al., 2020*; *Dal Santo et al., 2023*; *Bero, Chiu & Grundy, 2019*; *Boutron & Ravaud, 2018*; *Cruz Rivera et al., 2017*; *Levati et al., 2016*; *Albarqouni, Elessi & Abu-Rmeileh, 2018*; *Bentley et al., 2019*). This makes it difficult to summarize the methods and replicate them in future projects. This review can hopefully inspire researchers to use standardized methods in the future to improve reproducibility and comparability across research waste evaluation studies. Standardization promotes reproducibility and comparability, crucial for validating findings and enabling meaningful cross-study comparisons. Furthermore, standardized methods promote transparency and accountability in research practices. They offer a clear framework for assessing the quality and reliability of research findings, which is essential for upholding the integrity of scientific literature. By advocating for adherence to standardized methods, this scoping review seeks to enhance research quality and efficiency in biomedical sciences. We are currently focusing on systematic and scoping reviews because they comprehensively synthesize existing literature on research waste. However, it's important to acknowledge that other types of meta-research studies, such as cross-sectional designs and meta-epidemiological studies, can also offer valuable insights into the characteristics and extent of research waste. These studies can effectively use standardized tools to characterize and evaluate research waste. Including a broader range of study designs can improve our understanding and assessment of research waste across different types of research. It is favorable to consider study designs included in systematic reviews when evaluating research waste since the methods used to examine, *e.g.*, systematic reviews, should differ from those used for randomized controlled trials. There has been increased attention toward the problem, which is apparent by the increasing volume of reports in recent years (*Rosengaard et al., 2024b*). In the future, it could become standard to incorporate an evaluation of aspects of research waste into reporting guidelines for

systematic reviews, like the recommendation for assessment of risk of bias that is included in the current version of the PRISMA guideline (*Page et al., 2021*).

## CONCLUSIONS

In systematic reviews, scoping reviews, and overview reviews, the most examined aspects of research waste were the assessment of methodological quality and adherence to reporting guidelines. Many different tools and approaches were utilized, resulting in a diverse range of evaluation methods. This may present challenges in comparing examinations and performing meta-research in the future. This review guides researchers in selecting methodologies and contributes to the ongoing discourse on optimizing research efficiency.

### Funding
This work was supported by Tømmerhandler Johannes Fogs Fond (No. 2024-0134), and Grosserer L.F. Foghts Fond (No. 22.406). The funders had no role in study design, data collection and analysis, decision to publish, or preparation of the manuscript.

### Grant Disclosures
The following grant information was disclosed by the authors:
Tømmerhandler Johannes Fogs Fond: 2024-0134.
Grosserer L.F. Foghts Fond: 22.406.

### Competing Interests
The authors declare there are no competing interests.

### Author Contributions
- Louise Olsbro Rosengaard conceived and designed the experiments, performed the experiments, analyzed the data, prepared figures and/or tables, authored or reviewed drafts of the article, and approved the final draft.
- Mikkel Zola Andersen conceived and designed the experiments, analyzed the data, prepared figures and/or tables, authored or reviewed drafts of the article, and approved the final draft.
- Jacob Rosenberg conceived and designed the experiments, analyzed the data, prepared figures and/or tables, authored or reviewed drafts of the article, and approved the final draft.
- Siv Fonnes conceived and designed the experiments, analyzed the data, prepared figures and/or tables, authored or reviewed drafts of the article, and approved the final draft.

### Data Availability
The data is available at Zenodo: Rosengaard, L. O., Andersen, M. Z., Rosenberg, J., & Fonnes, S. (2024). Dataset to several methods for assessment of research waste: a scoping review [Data set]. Zenodo. https://doi.org/10.5281/zenodo.13824997.

## Supplemental Information

Supplemental information for this article can be found online at http://dx.doi.org/10.7717/peerj.18466#supplemental-information.

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
