# Peer review of "Several methods for assessing research waste in reviews with a systematic search: a scoping review"

_PeerJ, doi:10.7717/peerj.18466_

## Round 0.1 · original submission · Major Revisions

Thank you for your submission. As you can see, we have comments from two reviewers who have provided insightful and constructive comments about your manuscript. Reviewer #1's report raises some challenging questions that might require significant rework to address. Reviewer #2's report similarly raises questions that will require careful consideration and response. If you feel that you can address their collective remarks, I would be happy to reconsider a revised copy of your manuscript that indicates the revisions made along with point-by-point responses to each of the reviewers' points, indicating for each how the manuscript has been changed in response or why you believe that the particular comment does not require revisions.

Reviewer 1 ·

Basic reporting

English language is fine, references are fine.
Introduction is short and Discussion also - they did not provide many relevant references to the prior works about research waste.

Experimental design

Scoping review is a valid method, but there is no consensus definition about the methods to be used to assess research waste. Please see my detailed notes below.

Validity of the findings

Findings are dubious in the light of the unclarity of what the authors actually aimed to do. Please see my comments below.

Additional comments

Dear Editor,

I have reviewed the manuscript peerj-100923. This manuscript has an inherent problem that it is completely unclear whether the authors relied solely on analyzing reporting in the analyzed studies or they predetermined methods for assessing research waste. In the first case, they would extract only data from the studies that said, for example, “we used risk of bias assessment to analyze research waste”. In the second case, the authors would have a pre-determined list of methods that they think analyze research waste.

If the first scenario is true, that needs to be clearly reported in a way to restructure reporting in the entire manuscript.

If the second scenario is true (the authors made a list of pre-determined methods for assessing research waste), then the study design is completely flawed. In that case, the review “using systematic search” does not need to mention words “research waste”, but can still use methods that the authors consider to be assessment of research waste.
The authors wrote in the limitations “we only examined those reports that defined research waste as a problem”, but this is not reflected in the eligibility criteria written in the abstract or methods in the main text.

It is likely that there are also other methods to “assess research waste”, but they were not use in the very small number of included reviews in this study (n=93).

Also, to my best knowledge, there is no consensus definition of methods to assess research waste?

The authors refer to the MINUS definition of research waste, but the reference for this definition is manuscript “under review” from this group of authors. Thus, it appears that the authors created a new definition of research waste and conducted this study according to their definition, which could likely be an issue, because this is not an international consensus definition of research waste.


Abstract:
-To remove “before conducting our scoping review” and replace with: prospectively (and then provide a hyperlink to the OSF registration in the bracket after the word “prospectively”).
-The aim mentions systematic reviews, scoping reviews and overviews, but the study title mentions only systematic reviews; this is discrepant.
-“reviews with systematic searches” is the term that occurs late in the Abstract. It is not defined/mentioned before in the abstract.
-The aim should be reformulated into reporting that you analyzed reviews with systematic searches (including systematic reviews, scoping reviews and overviews) that examined and discussed research waste. This last section is also important “examined and discussed research waste”, because this is mentioned only in the Methods of the main text.
-Abstract mentions assessment of risk of bias, systematic review methodology (AMSTAR) and adherence to reporting guideline as methods for assessing research waste. This is very unclear. Did the authors of those studies explicitly report that they used those approaches to “assess research waste”?

-Reference #5 in the first sentence of Methods is difficult to understand, why it is supporting this sentence. This should be clarified.
-Reason for the 30-year limit on the search is unclear, and needs to be explained. The authors did mention it as a limitation, but the rationale for this decision in the Methods is not reported.
-“any method used to examine” should probably be revised into “any method reported to be used…”
-The authors did not define their meaning of “reviews with systematic searches” – this is very important because there is no consensus definition of a systematic review and no consensus definition of a universal “systematic search”.
-Did the authors of this study have a predefined definition of what are the methods to be used for “examining research waste”, or the authors simply anylzed what the authors of reviews reported as methods used for examining research waste?
-The exclusion criterion “reviews from non-biomedical sciences” is difficult to understand since PubMed and Embase were searched. Both information sources contain medical literature. It is unclear why would the authors exclude reviews of veterinary sciences – that field can also be prone to research waste.
-This is confusing “At least two reviewers independently screened” – can you please be specific how many authors independently screened each record?
-Data extraction was also conducted in Covidence?
-Extracted data: “methods used for investigating research waste” – was this supposed to be “methods reported by authors as used for investigating research waste”. Because if this is not the case, this means that you need to define what you considered as methods for investigating research waste. And in that case you would need to conduct a whole different study, not just look for reviews that have mentioned research waste, but look for any reviews with systematic search and whether they used those prespecified methods.

Results
-In Figure 1, multiple issues are confusing when reading the reasons for excluding full texts. The authors excluded 123 studies because “no definition of research waste”. But it is not reported in the study methods that the authors of eligible studies had to define research waste. Furthermore, the meaning of “concerning miscondunct only” is unclear. The authors wrote that they excluded 7 books, but did not write in the eligibility criteria that the books are excluded. The authors excluded 727 studies because “not a systematic search review”, but it is not clear how they defined systematic search. Finally, meaning of “No assessment of research waste” is unclear. By looking at this list of exclusion reasons, eligibility criteria for this study become very, very unclear.
-Additional supplementary file should be provided to list references of excluded studies and reasons for excluding them.
-The authors use discrepant reporting. The Results of the main text say “the reviews examined a median… studies for research waste”. The abstract says “the reviews included median 90 studies”. It is unclear whether the author use this as synonyms (included studies vs studies examined for research waste). This should not have the same meaning.
-“The most examined study designs” – is this supposed to be like this, or to write “the most frequently included study designs…”?
-The sentence “There has been an increased awarenesss…” is unclear and not supported with any numbers.
-The section “Research waste aspects and tools” mixes “aspects” and “tools”, and it is not clear what the authors defined as “aspects” and whether they pre-defined tools for assessing research waste.
-Figure 2: this is unclear: “pr.”
-Figure 3: it is unclear what is the difference between quality assessment and conduct examination. The term “conduct examination” is mentioned twice in the Results, but not anywhere else and not explained.

Discussion
-It is not clear what this means “performed a descriptive analysis…”.

-The authors write “This makes it difficult to summarize the methods and replicate them in future projects”. And, yet, they wrote in the abstract that “This review guides researchers in selecting methodologies…“. I do not think that this study can be used as such guide, as it had limited number of included studies, with dubious eligibility criteria.

Generally, English language is fine, references are fine.
Referenced literature: Introduction is short and Discussion also - they did not provide many relevant references to the prior works about research waste.

EBR: the authors did not mention the concept of evidence-based research (EBR) in terms of the research waste.

-In the Results, it is indicated “the majority were systematic reviews (n=73)”. On the contrary, in the Discussion, it is written: “as this scoping review only includes systematic reviews”. This is very confusing.

Data sharing: It is commendable that the authors shared their raw data extracted for this study, on Zenodo, to be publicly available. The authors should add a “Data availability statement” also in the end of their article, among Declarations.

Author contribution: The authors should clearly describe the contribution of all authors in detail, and not by using this generic sentence.

·

Basic reporting

OK

Experimental design

Correct design

Validity of the findings

Seems valid

Additional comments

Comments to
Several methods for assessing research waste in systematic reviews: a scoping review (#100923)

The authors should be praised for looking deeper into this important topic. However, the authors should consider these comments before publication is considered:


1. The authors are describing the main topic as “research waste”. I would suggest the authors to consider a more precise term “avoidable research waste” as some research waste is unavoidable and thus not important to assess.
2. The authors should in general consider that in some cases high risk of bias or questionable research practice may not always lead to research waste. Even studies with high risk of bias or otherwise performed in a questionable way may not be waste.
3. A reference to this statement would be helpful: “Research waste is present in all study designs” (line 55-6) and to this statement as well: “can have significant consequences for science, including reducing the reliability of research findings and contributing to the inefficient use of resources” (line 56)
4. The authors write “Systematic evaluations could help manage this rapid data growth and improve research quality.”, it could be helpful to know what to evaluate more precisely. (line 59-60)
5. The authors write: “Systematic reviews are considered the highest level of evidence in the field of research” (line 61). However, Cochrane does not state that, and I would suggest that the authors considered to explain that systematic reviews is the basic unit of knowledge translation or evidence (Tricco AC, Tetzlaff J, Moher D. The art and science of knowledge synthesis. Journal of Clinical Epidemiology. 2011;64(1):11-20) instead of highest level. One could argue that a systematic review is not better than the studies included – when it comes to the possible conclusion that can be made.
6. A reference to this statement would be helpful: “There is uncertainty in how research waste is assessed” (line 68).
7. I guess that there is some kind of a misunderstanding of two studies (one included (Sawin 2016) and one just referred to (Robinson 2011). Both are NOT systematic reviews, but cross-sectional meta-research studies that uses systematic reviews to identify the original studies to be included in their analyses. The benefit of using a systematic review is that other similar studies already are identified. Suggest considering excluding the Sawin study. (Line 128)
8. A reference to this statement would be helpful: “There has been an increasing awareness of methods for assessing research waste in the last ten years.” (line 133-4).
9. The Discussion is completely lacking (A) A discussion of the meaning or implications of the findings and (B) discussion of results in relation to results of earlier similar studies (or even better systematic rereview of earlier similar studies). If no studies having evaluated the objective of the present study, there is possible studies that are adjacent to the objective. (line 236-237)
10. I agree that the mentioned methods are a strength, but it is just standard methods, thus not really a strength. If not used it would be a limitation. (line 237-240)
11. Related to line 246-248 I come to think of a study we are preparing dealing with the problem of citation bias (also a scoping review). In our search we identified a number of studies that were not talking about citation bias, but how to increase citation rate by using approaches that normally would be classified as citation bias. Maybe that could be the case here as well? Just a thought.
12. I think it is too narrow only to talk about evidence syntheses (SR, ScR etc.). In many cases other kinds of meta-research studies (for example a random selection of studies in a cross-sectional design) would be just as helpful to evaluate the characteristics and size of research waste. (line 251-252)
13. A reference to this statement would be helpful: “There has been increased attention toward the problem, which is apparent by the increasing volume of reports in recent years” (line 259-260)

---

## Round 0.2 · Minor Revisions

Thank you for your revisions. One of our reviewers has provided six points that I feel all warrant being addressed (accepting and making revisions in light of or explaining why you do not feel changes are required for that particular point) in a revised version. Some of these are very simple and none look likely to require too much effort on your part. I look forward to seeing a revised version with a point-by-point response to these comments in due course.

Reviewer 1 ·

Basic reporting

It is fine.

Experimental design

It is fine.

Validity of the findings

Ok.

Additional comments

Dear Editor,

I have reviewed the Revision 1 of the manuscript peerj-100923.
The responses are very detailed and the manuscript has been much improved. Here are my additional comments:
1. The authors mentioned in the Response that they retrospectively aligned their methods with the published MINUS framework, but I do not see this mentioned in the revised manuscript?
2. “There has been an increasing awareness of methods…” – I would suggest to revise this sentence, and any mention of “awareness” because “awareness” was not measured.
3. I suggest to delete the sentence “Extending beyond 30 years would include outdated methods…” – this does not sound right. We do not know what would be outdated methods in this context. Research waste is very recent concept, anyway.
4. “Excluded reviews from non-biomedical sciences” – this needs to be revised for clarity in the manuscript as it does not sound correct. The response says “human health-related research” so the proper wording in the exclusion criteria instead of “reviews from non-biomedical sciences” would be: “reviews that did not analyze health-related research on humans”.
5. The authors wrote “we included published reviews…” – but books are also published. So I would suggest to clarify in the eligibility criteria that you restricted your sample to reports published in scholarly journals. I would not mention “peer-reviewed” journals because that would require checking whether the jornals (each and every one) really used peer-review.
6. Lack of the list of excluded studies – since the authors used Covidence, this list could be easily retrieved from Covidence. I am not sure exactly why the authors cannot provide this list.

Sincere regards

---

## Round 0.3 · accepted · Accept

Thank you for your revisions and rebuttal. There are no further comments from our reviewers and I am delighted to accept your manuscript. Well done!